# Coordination Among Neural Modules Through a Shared Global Workspace

Anirudh Goyal [1], Aniket Didolkar[1], Alex Lamb [5], Kartikeya Badola [6], Nan Rosemary Ke [2],
Nasim Rahaman [1,3], Jonathan Binas [1], Charles Blundell [2], Michael Mozer [4], Yoshua Bengio [1]

## Abstract

Deep learning has seen a movement away from representing examples with a monolithic hidden state towards a richly structured state. For example, Transformers segment by position, and object-centric architectures decompose images into entities. In all these architectures, interactions between different elements are modeled via pairwise interactions: Transformers make use of self-attention to incorporate information from other positions and object-centric architectures make use of graph neural networks to model interactions among entities. We consider how to improve on pairwise interactions in terms of global coordination and a coherent, integrated representation that can be used for downstream tasks. In cognitive science, a *global workspace* architecture has been proposed in which functionally specialized components share information through a common, bandwidth-limited communication channel. We explore the use of such a communication channel in the context of deep learning for modeling the structure of complex environments. The proposed method includes a shared workspace through which communication among different specialist modules takes place but due to limits on the communication bandwidth, specialist modules must compete for access. We show that capacity limitations have a rational basis in that (1) they encourage specialization and compositionality and (2) they facilitate the synchronization of otherwise independent specialists.

## 1 Introduction

Deep Learning has seen a movement towards more structured models with cleaner separation between different pieces of information often handled by different components. The induced structure, and separation of knowledge has improved generalization, model-size scaling, and long-range dependencies (Berner et al., 2019; Vinyals et al., 2019; Brown et al., 2020). This opens up questions about how to achieve coherence and coordination between different components in such architectures. Looking back to the 1980s, the focus in AI was much less on learning and more on constructing articulated, multi-component architectures and examining how intelligence might emerge from interactions among this collection of simple, functionally specialized components (Fodor, 1983; Braitenberg, 1986; Minsky, 1988; Brooks, 1991). Each

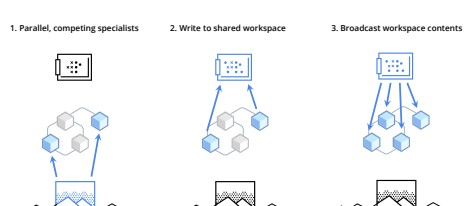

Figure 1: **Step 1:** an ensemble of specialist modules doing their own default processing; at a particular computational stage, depending upon the input, a subset of the specialists becomes active. **Step 2:** the active specialists get to write information in a shared global workspace. **Step 3:** the contents of the workspace are broadcast to all specialists.

of these specialist modules is on the scale of a typical component of a computer program, like a subroutine that implements a narrow, prespecified function from certain input contents to certain

[1] Mila, University of Montreal, [2] Google Deepmind, [3] Max Planck Institute Germany, [4] Google Research, Brain Team, [5] Microsoft Research, New York, NY, [6] Indian Institute of Technology, Delhi, Corresponding authors: `anirudhgoyal9119@gmail.com`

output contents. Through appropriate communication and coordination, a set of specialists can achieve complex, dynamic, and flexible behavior patterns.

As a concrete illustration, consider the task of driving a car in terms of specialists. One specialist might monitor the position of the car with respect to lines on the road, and another specialist might adjust the steering direction based on the perceptual data. In addition, there might be specialists which provide alerts when certain events occur, such as loud sounds, reaching a critical intersection on a route, or coming into close proximity to the car in front. To execute the task of driving the car properly, all these specialists need to interact coherently and broadcast their individual information to each other.

Arguably, modern ML and AI has yet to develop broad architectural frameworks for learning both the specialist modules and how they should *interact*, while the classical view lacks an articulate story about how learning could take place successfully in such frameworks. In this article, we revisit this classical view with modern machine learning tools based on end-to-end learning and differentiable memory and attention mechanisms. Inspired by the Global Workspace Theory (Baars, 1993; Dehaene et al., 1998; Shanahan and Baars, 2005; Shanahan, 2006; 2010; 2012; Dehaene et al., 2017) from cognitive neuroscience, we argue that more flexibility and generalization emerge through an architecture of specialists if their training encourages them to communicate effectively with one another via the bottleneck of a shared workspace (Figure. 1).

**Distributed specialist modules.** From a computational perspective, articulated multi-component architectures composed of sparsely interacting specialist modules show desirable scaling properties (e.g., more specialists can seamlessly be added), increased robustness (the system can tolerate the removal of or changes in individual specialists), and efficiency (information is processed predominantly locally, reducing the cost of communication between specialists). However, modularization also requires mechanisms to establish sharing of compatible representations across specialists, a form of shared internal language. While portions of a task might be solved by independent specialists, synchronization is critical particularly when there are statistical, functional, or causal dependencies among the specialists.

**Coherence through a shared workspace.** In cognitive neuroscience, the Global Workspace Theory (GWT) (Baars, 1993; Dehaene et al., 2017) suggests an architecture allowing specialist modules to interact. The key claim of GWT is the existence of a shared representation—sometimes called a blackboard, sometimes a workspace—that can be modified by any specialist and that is broadcast to all specialists, along with the notion that write access is limited to maintain coherence. Our interpretation of this restriction on write access is that it stems from an assumption on the form of the joint distribution between high-level concepts. In this paper, we explore a communication and coordination scheme similar to the one proposed by GWT for modern neural network architectures like Transformers (Vaswani et al., 2017; Dehghani et al., 2018; Parmar et al., 2018; Radford et al., 2019; Brown et al., 2020) and attention-based modular architectures (Goyal et al., 2019; Rahaman et al., 2020; Mittal et al., 2020a; Goyal et al., 2020; Madan et al., 2021).

In terms of our driving example, the workspace could be used to override default behaviors by giving high priority to specialist modules which provide alerts of various sorts (loud sounds, presence of a child on the street), allowing specialists which respond to such alerts to take control of behavior over default driving routines. This scenario implies that prioritization of signals in a shared workspace is critical.

**A shared communication channel necessitates common representations.** For a multitude of specialist modules to cooperate, a common language is necessary (Baars, 1997). For example, in the driving scenario, alerts may come from auditory or visual processing specialists, but regardless of the source, a signal for danger must be placed in the workspace to override default behavior, whether that behavior is controlled by a radio-tuning specialist or a steering specialist. Although specialist modules can be pre-wired to have compatible communication interfaces, we will model an architecture in which an ensemble of specialist modules is *trained in coordination*, which should lead to a shared language (Colagrosso and Mozer, 2005). Internally, individual specialists can use whatever form of representations that serves them, but their inputs and outputs require alignment with other specialists in order to synchronize. For example, an unusual event such as a rough thud under the wheels might not have been previously experienced, but the mere signalling of novelty

---

In the literature, specialists are sometimes referred to as processes or agents.

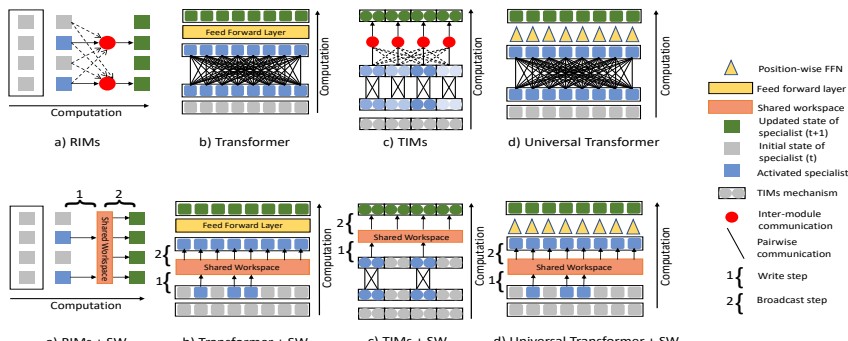

Figure 2: **Using a Shared Workspace for creating global coherence in RIMs, Transformers, TIMs and Universal Transformers (UT).** (Top Half) All four of these architectures use pairwise communication (using key-value attention) to establish coherence between individual specialist modules. In the case of RIMs (Goyal et al., 2019) and TIMs (Lamb et al., 2021), these specialists are independent modules that compete with each other in order to take control over the state update based on a given input. In the case of Transformers (Vaswani et al., 2017) and Universal Transformers (Dehghani et al., 2018), each specialist is associated with a different position. Activated specialists are denoted by a blue shade and the intensity depends on the degree of activation. In the case of Universal Transformers, the state update dynamics for each position is shared across all layers and all positions (denoted by a yellow triangle). (Bottom Half) We replace pairwise communication with a shared workspace to create *global coherence* between different specialists. Communication using the shared workspace is a two-step process (as denoted by 1 and 2 in the figures). In the first step (1), specialists compete for write access to the shared workspace, resulting in a subset of them being activated (in blue), and only the activated specialists perform the write operation on the workspace. In the second step (2), the contents of the shared workspace are broadcast to all the specialists.

could override default specialists. Without a global communication channel, specialists would have to learn to communicate through pairwise interactions, which might limit coordination of behavior in novel situations: global communication ensures exchangeability of knowledge to achieve systematic generalization.

## 2 SYNCHRONIZING NEURAL MODULES THROUGH A SHARED WORKSPACE

We investigate a neural architecture reminiscent of the GW model, where a number of sparsely communicating specialist modules interact via a shared working memory. In particular, we extend the Transformer (Vaswani et al., 2017), attention and slot-based modular architectures (Goyal et al., 2019) by adding a shared workspace and allowing modules (each representing an entity) to compete for write access in each computational stage.

**Key-value attention.** Key-value attention defines the backbone of updates to the hidden states in the proposed model. This form of attention is widely used in self-attention models and performs well on a wide array of tasks (Bahdanau et al., 2014; Vaswani et al., 2017; Santoro et al., 2018). Key-value attention selects an input value based on the match of a query vector to a key vector associated with each value. To allow differentiability and thus easier learnability, selection is soft and computes a convex combination of all the values. Such a mechanism makes it possible to change on-the-fly both the source of input and how the shared workspace is updated. It also makes the outputs of the specialists and the elements of the memory permutation invariant: they should be considered as an *unordered* set of elements to be selected by an attention mechanism from the contents of specialists. More precisely, soft attention uses the product of a *query* (represented as a matrix $Q$ of dimensionality $N_r \times d$, with $N_r$ queries, and $d$ the dimension of each query) with a set of $N_o$ objects each associated with a *key* as a row in matrix $K^T$ ($N_o \times d$). After normalization with a softmax the resulting convex weights are used to combine the *values* $V_i$ (row $i$ of matrix $V$): where the softmax is applied to each row of its argument matrix, yielding a set of convex weights. For our experiments, we use multihead dot product attention.

**Neural modules with pairwise interactions.** Our approach to synchronizing neural modules is highly general and mostly agnostic to the task, domain, or specific choice of architecture, with the

only requirement being that the model consists of multiple specialist modules which either operate independently or have sparse interactions requiring to only match pairs of modules at a time. Our goal is to explore how introducing a shared workspace can help these modules to become better synchronized and coordinated. We show the utility of the shared workspace for synchronization in (a) Transformers (Vaswani et al., 2017), in which all interactions between positions are performed via attention, and (b) slot-based architectures like Recurrent Independent Mechanisms or RIMs (Goyal et al., 2019) in which all pairwise interactions between modules are performed via attention. In the context of slot-based architectures, each slot's content is associated with a specialist module, whereas in Transformers different entities each associated with a different position acts as a specialist module (Figure 2).

Both Transformers and RIMs utilize a self-attention mechanism for sharing information between modules, typically implemented in a pairwise manner, i.e., each specialist attends to every other specialist. Instead, we facilitate information sharing among specialist modules through a *limited capacity shared workspace*. In this framework at each computational stage, different specialists compete for write access to the common workspace. The contents of the workspace, in turn, are broadcast to all specialist modules simultaneously.

**Notation.** The input is processed through a sequence of computational stages indexed by $t$, and at each stage, $n_s$ entities are operated on (i.e., $n_s$ different modules in slot-based architectures like RIMs or $n_s$ different positions in the case of Transformers). Each of these $n_s$ specialist modules has a distinct internal $n_h$-dimensional state $\boldsymbol{h}_t^k$, for $k \in \{1, ..., n_s\}$. The specialist modules communicate with each other via a shared workspace divided into $n_m$ memory *slots*, each consisting of a vector of $n_l$ elements, denoted $\boldsymbol{M} = [\boldsymbol{m}_1; \dots \boldsymbol{m}_j; \dots \boldsymbol{m}_{n_m}]$. The shared workspace is updated across different computational stages i.e., different time-steps in recurrent architecture and different layers in the case of Transformers. At each computational stage $t$, different specialists compete for writing in the shared workspace, but all specialists can read from the current state of the workspace. In the case of an autoregressive task, we can restrict the information sharing to previous positions and keep a separate version of the workspace for each position.

## 2.1 Specifics of the Shared Workspace.

***Step 1: Process Input to obtain an entity representation for each specialist.*** The first step is external to the proposed method, and involves processing the input to form the initial representation vector for each of the different specialists. Different common deep learning architectures can be used to form the representation of different specialists. For example, Transformers start with a matrix $n_s \times n_h$ whose rows are initialized as the $n_h$-dimensional embeddings of the input at each position of the sequence. Slot-Based Recurrent architectures like RIMs consist of a single-layer recurrent structure where the hidden state $\mathbf{h}_t$ at computational stage $t$ is decomposed into the substates of the $n_s$ specialists, $\mathbf{h}_t^k$ for $k = 1, ...n_s$.

In the proposed scheme, within each computational stage, the updates of the hidden state of different specialists follow a two-step process. First, specialists compete and write to a shared workspace. Second, information from the workspace gets broadcast to all the specialists, as detailed next.

***Step 2: Writing Information in the shared workspace.*** The specialists compete to write into the shared workspace, whose contents need to be updated in the context of new information received from different specialists. This step ensures that only the critically important signals make it to the shared workspace, therefore preventing the workspace from being cluttered. Let matrix $\boldsymbol{R}$ represent the combined state of all the specialists (i.e. $h_t^k \quad \forall k \in \{1, \dots, n_s\}$ as the rows of $\boldsymbol{R}$). In order to implement the competition between specialists to write into the workspace, we use a key-query-value attention mechanism. In this case, the query is a function of the state of the current workspace memory content, represented by matrix $\boldsymbol{M}$ (with one row per slot of the memory), i.e $\widetilde{\boldsymbol{Q}} = \boldsymbol{M}\widetilde{\boldsymbol{W}}^q$. Keys and values are a function of the information from the specialists i.e., a function of $\boldsymbol{R}$. We apply dot product attention to get the updated memory matrix: $\boldsymbol{M} \leftarrow \text{softmax}\left(\frac{\widetilde{\boldsymbol{Q}}(\boldsymbol{R}\widetilde{\boldsymbol{W}}^e)^{\mathrm{T}}}{\sqrt{d_e}}\right) \boldsymbol{R}\widetilde{\boldsymbol{W}}^v$. The use of a regular softmax to write into $\boldsymbol{M}$ leads to a standard soft competition among different specialists to write in the shared workspace. One can also use a top-$k$ softmax (Ke et al., 2018) to select a fixed number of specialists allowed to write in the shared workspace: based on the pre-softmax values, a fixed number of $k$ specialists which have the highest values are selected, and get access to write in the shared workspace. Selection with a top-$k$ softmax is a hybrid between hard and soft selection. We

denote the set of thus selected specialists as $\mathcal{F}_t$. We note that we can apply the attention mechanism multiple times to distill information from different specialists into the shared workspace. Here, the contents of the shared workspace are updated in the gated way as proposed in RMC (Santoro et al., 2018). We ask the reader to refer to appendix section C for more details.

***Step 3: Broadcast of information from the shared workspace.*** Each specialist then updates its state using the information broadcast from the shared workspace. We again utilize an attention mechanism to perform this consolidation. All the specialists create queries $\widehat{q}_k = h_t^k \widehat{W}^q$, which are matched with the keys $\widehat{\kappa}_j = (m_j \widehat{W}^e)^{\mathrm{T}} \quad \forall k \in \{1, \ldots, n_s\}, \ j \in \{1, \ldots, n_m\}$ from the updated memory slots, forming attention weights $s_{k,j} = \mathrm{softmax}\left(\frac{\widehat{q}_k \widehat{\kappa}_j}{\sqrt{d_e}}\right)$. The memory slot values generated by each slot of the shared workspace and the attention weights are then used to update the state of all the specialists: $h_t^k \leftarrow h_t^k + \sum_j s_{k,j} \widehat{v}_j$ where $\widehat{v}_j = m_j \widehat{W}^v \quad \forall k \in \{1, \ldots, n_s\}$. After receiving the broadcast information from the workspace, each specialist update their state by applying some dynamics function i.e., one step update of LSTM or GRU units in the case of recurrent architectures, and a feedforward layer in the case of Transformers. This yields the new value $h_{t+1}^k$ for the $k$-th specialist, from which we start the next stage $(t + 1)$.

Replacing pairwise interactions among neural modules with interaction facilitated by the shared workspace allows for the following:

***1. Higher-order (HO) interaction among neural modules.*** The two-step write-read process first allows each memory slot to store a 'filtered summary' of the current input where the 'filter' is determined by the previous state of that slot ('Query' for the write step). Neural modules then summarize the information contained in these slots and update their state. Hence unlike pairwise interaction, messages passed among neural modules in the shared workspace setting also include HO interaction terms; those consisting of more than 2 modules at a time. Naturally, HO interaction require that messages passed among neural modules lie in the same representation space, which is precisely what we aim to achieve by allowing message passing only via a singular global channel.

***2. Dynamic filtering due to persistence of memory.*** With a shared workspace (SW), contents of the memory slot play a key role in filtering and summarizing the information contained in the input at a given time step. Persistence of memory throughout an episode 1) would allow the memory layer to summarize and filter information based on what it has seen thus far 2) should ideally lead to better generalization as the model is able to dynamically modify its filtering machinery for a particular input. In contrast, "inducing points" in Set Transformers (Lee et al., 2019) are fixed after training and hence the bottleneck cannot adjust itself on the fly for any new input. We present comparisons on several tasks in section 4. They show the importance of these two properties by comparing performance of SW with a) $2\times$Self-Attention (to simulate HO interaction without global communication) b) a version without memory persistence, in Appendix D.

***Computational Complexity of using shared workspace for synchronizing different specialists.*** To encourage a coherent global coordination, Transformers and slot-based recurrent architectures rely on pairwise interactions captured via an attention mechanism. Unfortunately, such attention mechanisms scale quadratically with the number of specialists. Here, we propose a method which uses a shared workspace to create global coherence between different specialists and in the process, replaces the pairwise interactions of conventional dot-product attention. The computational complexity of the proposed method is thus *linear* in the number of specialists. In our experimentation, the number of memory slots is practically constant, which suggests a very favourable scaling behavior, and certainly much less than quadratic. As a point of reference, what would correspond to the number of slots in human working memory (Baars, 1993) is indeed very small (less than 10 slots).

## 3 RELATED WORK

This work taps into a line of reasoning put forward by historical works, such as Minsky (1988); Braitenberg (1986); Fodor (1983), wherein it is argued that in order to be able to deal with a wide spectrum of conditions and tasks, an intelligent system should be comprised of many interacting specialized modules or programs, rather than a single "one-size-fits-all" entity. While modular architectures have been the subject of a number of research directions, (Jacobs et al., 1991; Bottou and Gallinari, 1991; Ronco et al., 1997; Reed and De Freitas, 2015; Andreas et al., 2016; Rosenbaum et al., 2017; Fernando et al., 2017; Shazeer et al., 2017; Rosenbaum et al., 2019; Goyal and Bengio,

2020), we focus here on a mechanism for achieving coherence and synchronization between specialist modules via a global workspace shared between all specialists.

Prior works have explored incorporating slot-based memory in the context of recurrent neural networks (Graves et al., 2014; 2016; Santoro et al., 2018). In the context of transformers, Burtsev and Sapunov (2020) introduce *memory tokens* that are processed in addition to sequence tokens, whereas Dai et al. (2019) (Transformer-XL) propose to partition a long sequence to smaller segments and use the activations of the previous segment in memory while processing the current segment. Building on the latter, Rae et al. (2019) propose to store activations from prior segments in a compressed memory. However, these methods do not restrict memory writes to be sparse and competitive. Recent advances in this direction include the global neuronal workspace (GNW) model (Dehaene and Changeux, 2011), which identifies the global workspace with a large network of excitatory pyramidal neurons with long-range axonal processes connecting prefrontal and parietal cortices. Further, deploying a shared workspace to establish coherence between different specialists as opposed to using all-pair communication has an added benefit, in that it allows us to tackle the $O(n^2)$ complexity of self-attention. This makes our work related to previous work on reducing the computational complexity of dot product attention in Transformers. Lee et al. (2019) introduce the *ISAB* module, which maps between sets and comprises two dot-product attention layers. In the first layer, a set of trainable parameters are used as queries and the elements of the input set as keys; in the second layer, the output of the first layer is used as keys and the input set as queries. However, unlike in this work, the intermediate states (corresponding to the output of the first layer) are not maintained across layers. Concurrent to our work, (Jaegle et al., 2021) also introduced the idea of using a latent bottleneck for addressing quadratic complexity by learning a bottleneck but there are important differences. For example. in Perceiver the latent bottleneck iteratively queries the information about different positions, and does not maintain the representation of the different specialists. More precisely, in our proposed method different specialists write information in the workspace and then information gets read from the shared workspace. In Perceiver, the latent bottleneck iteratively reads information from the set of positions. We also show the applicability of the proposed idea both for slot based models and Transformers.

The proposed model can also be seen as integrating out different ideas popular in modular architectures (Andreas et al., 2016; Goyal et al., 2019), memory networks (Graves et al., 2014; Santoro et al., 2018) and mixture of experts (Jacobs et al., 1991), and hence combining some of their benefits in a unified architecture. The proposed model is factored as a set of specialists (incorporating modularity). The proposed model achieves coordination among different specialists via the use of a shared workspace (in the Neural Turing machines, there is only a single specialist i.e., without any modularity). Multiple experts can be active at the same time (generally not the case with a mixture of experts).

## 4 EXPERIMENTS

Here we briefly outline the tasks on which we applied the idea of the shared workspace and direct the reader to the appendix for some more experiments (Appendix G), full details on each task and details on hyperparameter settings for the model. The experiments have the following goals: (a) Demonstrate that the use of the shared workspace can improve results on a wide array of challenging benchmark tasks, with the goal of demonstrating the practical utility and breadth of the technique. (b) Show that the shared workspace addresses coherence between different specialists by achieving improved performance without requiring all pairwise interactions. Finally, to show wide applicability of our model, we integrate SW in TIMs (Lamb et al., 2021), SCOFF (Goyal et al., 2020) and BRIMs (Mittal et al., 2020b) and show improvements over the default communication method used in each.

**Making sense of the visual input.**   Using a shared workspace introduces a bottleneck in sharing of information between specialists. Since the size of the workspace is limited and generally much lower than the number of specialists, there is a limit to the amount of information that can be exchanged among specialists. We hypothesize that mediating communication through a limited capacity workspace should encourage the model to look at relevant information that is important for the downstream objective. We test this hypothesis on a set of visually challenging benchmarks. For our experiments, we use either Transformers or RIMs as a backbone. We consider variants of Transformers based on different subsets of important properties. *Transformers* [TR]: Self-attention based multi-layer architecture (Vaswani et al., 2017) with shared parameters across layers. *Set transformer* [ISAB]: Transformers where self attention is replaced by ISAB module (Lee et al., 2019). *Sparse*

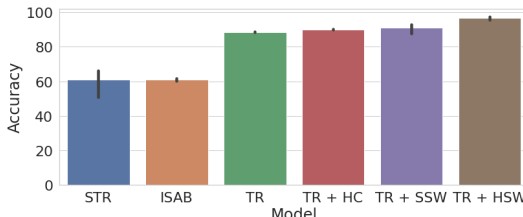

Figure 3: **Detecting Equilateral Triangles.** Here, we compare the performance of the Transformers with shared workspace to other Transformer baselines. Here, we plot the test accuracy for each model.

| Model | Top-1 % | Top-5 % |
|---|---|---|
| ISAB | $65.3_{\pm 0.025}$ | $83.6_{\pm 0.011}$ |
| STR | $70.6_{\pm 0.08}$ | $87.33_{\pm 0.06}$ |
| TR | $70.83_{\pm 0.44}$ | $87.8_{\pm 0.08}$ |
| TR + HC | $70.17_{\pm 0.31}$ | $88.33_{\pm 0.2}$ |
| TR + HSW (OURS) | $71.07_{\pm 0.04}$ | $88.6_{\pm 0.49}$ |
| TR + SSW (OURS) | $71.33_{\pm 0.34}$ | $88.3_{\pm 0.05}$ |

Table 1: **Comparison on CATER Object Tracking**. Here, we compare the Top-1 and Top-5 accuracy of Transformers with shared workspace and Transformers with self-attention. We can see that Transformers with a shared workspace outperform those with pairwise self-attention.

*Transformers* [STR]: Transformers with sparse factorizations of the attention matrix (Child et al., 2019). *High Capacity Transformers* [TR+HC]: Same as TR but with different parameters across layers. *Transformers with Shared Workspace with soft-competition* [TR+SSW]: Transformers with different positions competing with each other to write in shared workspace using soft-competition. *Transformers with Shared Workspace with top-$k$ competition* [TR+HSW]: Transformers with different positions competing with each other to write in shared workspace using top-$k$ competition. For a more detailed description of all the tasks described below, we ask the reader to appendix section E.

**Detecting Equilateral Triangles**. We first use a simple toy task to test our hypothesis where the model should detect equilateral triangles in images (Ahmad and Omohundro, 2009). Each image is of size $64 \times 64$ and contains 3 randomly placed clusters of points. For equilateral triangles, the midpoints of these clusters are equidistant from each other. This is a binary classification task where the model has to predict whether the three given clusters form an equilateral triangle or not. To feed an image into a Transformer, we follow the same methodology as used in vision Transformers (Dosovitskiy et al., 2020). We first divide an image into equal sized $4 \times 4$ patches and treat each patch as a different input position of the Transformer.

To solve this task correctly, the model only needs to attend to relevant information i.e., to patches that contain the cluster of points. Therefore, using a limited capacity shared workspace should be useful here. Our results (presented in Figure 3) confirm this hypothesis. We can see that *Transformers with shared workspace attention converge much faster and reach higher accuracy as compared to the baseline Transformer*. Our method also outperforms Set Transformer by a significant margin.

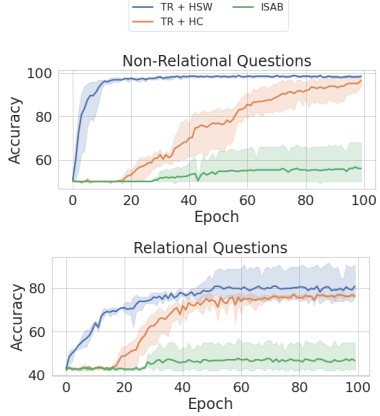

Figure 4: **Comparison on Sort-of-CLEVR relational reasoning**. Speed of convergence for relational and non-relational questions in the sort-of-clevr dataset. We can see that the proposed model converges much faster than the baselines in both cases.

**Multi MNIST Generation.** In this task, we train an Image Transformer (Parmar et al., 2018) (pixel-by-pixel, raster-order generative model) for next-pixel prediction on the "MultiMNIST dataset" where each image consists of 4 independently sampled MNIST digits stacked horizontally to form one image (see Figure 10 for demonstration). The main aim of this task is to observe the inductive biases that allow for specialization of mechanisms in TIMs (Lamb et al., 2021). Each image in the MultiMNIST dataset can be broken down into different sets of independent spatial components. Since the digits which make up the image are independently selected, the joint distribution of pixel intensities in any one of the four sections of the image is statistically independent of the pixel intensities in any other section of the image. Moreover each section of the image can be further broken down into independent spatial components: one that pertains to the background and one that pertains to the foreground. One can expect that architectures that are made up of sparsely interacting

different mechanisms to naturally capture this statistical independence by dividing labour among different mechanisms. While, for monolithic architectures, a major portion of their training time will be spent in learning these statistical independencies from scratch. We find that replacing the pairwise communication in TIMs with a shared workspace (TIMs + SW) leads to better and more interpretable division of labor among specialists as shown in Figure 5. From the figure, It is clear that the TIMs model is unable to divide labour among specialists with mechanism 2 being activation for all the pixels in the image. On the other hand, we can see that TIMs + SW is able to divide labor among specialists with each mechanism focusing on a different aspect of the image. We can see that mechanism 2 gets activated for the digits which are present towards the centre of each of the 4 columns while mechanisms 3 and 4 cover the background of the digits, with mechanism 3 covering the area between adjacent digits and mechanism 4 covering the area above and below the digits. Thus, we can see that using a shared workspace aids the division of labor among different specialists. We also find that TIMs + SW results in the least cross-entropy loss in the test set when compared to TIMs and Image Transformers (Parmar et al., 2018). Results shown in appendix Table 5.

**CATER: Object Tracking.** Cater is a spatio-temporal reasoning video dataset introduced in Girdhar and Ramanan (2019). Each video contains 3D objects organized in a $6 \times 6$ grid. Each object affords certain actions that can be performed on them. These actions result in movement of the concerned objects and change in their positions. Some of these actions include: *rotate*, *pick-place*, *slide*, *contain*. Throughout the duration of the video, a number of these actions are performed to get the final state of the grid. Note that only a single object undergoes an action, at any instant. The task that we focus on here is called *localization*. In this task, the goal is to predict the location of the target object in the final frame. In this case the target object is called a snitch. The snitch as well as the other objects move across the $6 \times 6$ grid. In some scenarios, the snitch may be covered by other objects hence hiding it from the view. In such cases, tracking the movement of the snitch across frames becomes essential. Therefore, capturing long-range temporal dependencies is essential to solve this task.

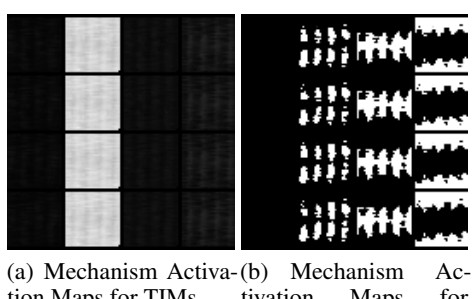

(a) Mechanism Activation Maps for TIMs

(b) Mechanism Activation Maps for TIMs+SW

Figure 5: This figure shows the mechanism activation map for all 4 mechanims used in the multimnist generation task for both TIMs and TIMs + SW. Both the images in the figure correspond to the activation maps from 4 different examples. Each activation map contains 4 mechanisms shown from left to right in a single row. Each mechanism is shown using a 32 x 32 image, a particular pixel in a mechanism activation map is shown in white if that mechanism was used during the generation of that pixel while generating the image.

The information exchange limit enforced by the limited capacity of the shared workspace should be useful here as well. For CATER, in some frames the snitch is not visible as it is covered by other objects. Therefore, ideally the model only needs to attend to frames in which the snitch is visible. Additionally, if the snitch is visible throughout the video in all frames, then to accurately predict the final position of the snitch, the model only needs to attend to the final frame of the video and can completely ignore the initial frames. The results for this task are presented in Table 1. We also experimented with both soft competition TR+SSW and hard competition TR+HSW, with only $k = 5$ specialists writing into the shared workspace. We can see that models with a shared workspace outperform those with pairwise multihead attention thus confirming our hypothesis about the benefits of a shared workspace for this task. As shown in Table 1 proposed method convincingly outperforms the Set Transformer.

**Relational Reasoning : Sort-of-CLEVR.** In relational reasoning, the model is tasked with answering questions about certain properties of various objects and their relations with other objects. The model is presented with an image and a question for that image. This task has a clear sparse structure as in order to answer the questions correctly, it needs to only reason about a specific subset of objects that the question mentions. For this task, we use the Sort-of-CLEVR dataset (Santoro et al., 2017).

Each image in Sort-of-CLEVR is of size $75 \times 75$ and contains 6 randomly placed geometrical shapes of 6 possible colors and 2 possible shapes. Each image comes with 10 relational questions and 10 non-

relational questions. Non-relational questions only consider properties of individual objects. On the other hand, relational questions consider relations among multiple objects. For more details about the question see appendix Figure 8. The input to the model consists of the image and the corresponding question. We first obtain a sequence of equal-sized patches for the image as in vision Transformers (Dosovitskiy et al., 2020). We concatenate the resulting patch sequence with the representation of the question and pass the combined sequence through the Transformer. Sort-of-CLEVR has a finite number of possible answers, hence this task is setup as a classification task.

We present the results for this task in Figure 4. We observe that *the Transformers with the shared workspace converge faster and outperform the baselines for relational as well as non-relational questions*. The superior performance with shared memory can be attributed to the inherent sparsity of this task. For instance, in non-relational questions, the model only needs to attend to a single object referenced in the question to answer it correctly, while relational questions only consider a small subset of objects in the image, thus sparsity is helpful for both these types of questions. Therefore, the limited capacity of the shared workspace forces the model to attend to only relevant information.

| Model | Num. Slots | ARI $\uparrow$ | MSE $\downarrow$ |
|---|---|---|---|
| SCOFF | - | $0.276_{\pm 0.001}$ | $0.083_{\pm 0.0}$ |
| SCOFF + SW | 2 | $0.154_{\pm 0.007}$ | $0.135_{\pm 0.002}$ |
| SCOFF + SW | 4 | $0.487_{\pm 0.085}$ | $0.059_{\pm 0.0}$ |
| SCOFF + SW | 5 | $0.915_{\pm 0.0}$ | $0.035_{\pm 0.0}$ |
| SCOFF + SW | 8 | $0.891_{\pm 0.001}$ | $0.039_{\pm 0.0}$ |
| SCOFF + SW | 10 | $0.351_{\pm 0.001}$ | $0.08_{\pm 0.0}$ |

Table 2: Here we show the performance of SCOFF augmented with shared workspace attention on the bouncing balls task. We also analyse the effect of varying number of slots in the shared workspace. This also shows that by increasing the number of slots performance decreases hence validating claims regarding bandwidth limited communication channel via shared workspace.

**Shared Workspace for Physical Reasoning.** In this task, we consider a set of bouncing balls and the model is tasked with predicting the trajectory of the balls at each step. In order to solve this task, a coherent picture of where and which objects will collide needs to be established by the learner. We use the bouncing-ball dataset from Van Steenkiste et al. (2018). We train the model for next-step prediction. We compare the proposed approach against SCOFF (Goyal et al., 2020). The results of our comparison are shown in Table 2. We use the ARI and MSE metric for comparison. ARI measures how well the different balls are segregated into different slots, higher ARI means better segregation. We can see that using a shared workspace results in higher ARI as compared to pairwise communication in SCOFF. Thus, using a shared workspace results in better division of labor among specialists. We also compare the proposed method against other baselines in appendix section F.1.

**Shared Workspace for Atari Video Games.** We start by training RIMs, RIMs + shared workspace (SW) on three "source" games (Pong, River Raid, and Seaquest) and test if the learned features transfer to a different subset of randomly selected "target" games (Alien, Asterix, Boxing, Centipede, Gopher, Hero, James Bond, Krull, Robotank, Road Runner, Star Gunner, and Wizard of Wor). We take a sufficient number of specialists in RIMs (10). We train on source games for 10M steps, and then fine-tune on transfer games for 10M more steps. We choose these games as they were also used in the original RIMs paper (Goyal et al., 2019). Using a suite of 36 game pairs, we find that RIMs + SW outperforms RIMs on both game A (a median performance ratio of 1.13; mean of 1.16) and game B (a median performance ratio of 1.11; mean of 1.15). The improved performance with RIMs + SW is due to better forward transfer (knowledge acquired for game A facilitates the learning of game B) and reduced backward interference (knowledge acquired for game B does not disrupt knowledge acquired for game A), presumably thanks to a more appropriate modularization of knowledge.

## 5 CONCLUSION

Inspired by cognitive neuroscience global workspace theories, we have proposed a shared workspace model for establishing coherence among modular neural specialists while exchanging information in a systematic way. We show that using a limited capacity shared workspace as a bottleneck for mediating communication among specialists results in better performance across a wide range of visual reasoning benchmarks as compared to the pairwise interactions typically used in self-attention schemes. The proposed approach combines several key properties: knowledge and expertise is divided among specialists, they compete to post new contents to the workspace, and after being updated, the shared workspace is accessible to all specialists for their own updates.

ETHICS STATEMENT

The authors do not foresee any negative social impacts of this work, but of course the accumulation of improvements in ML could be misused as it may give more power to nefarious agents.

REPRODUCIBILITY STATEMENT

We use Algorithms 1 and 2 for our experiments, we will be releasing the code after the review process. We also provide our code in the supplementary material.

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

# Part I

# Appendix

## A   PSEUDO CODES

Alg. 1 shows the integration of shared workspace with RIMs (Goyal et al., 2019). We replace the direct module to module interaction via attention in RIMs, with shared workspace. Specialists compete to write in the shared workspace, and the contents of the workspace are broadcasted to all the specialists.

Alg. 2 shows the integration of the shared workspace with TIMs (Lamb et al., 2021). Again we replace the direct module to module communication in TIMs, with a shared workspace.

---

**Algorithm 1:**  Shared Workspace integration with RIMs

---

***Input:*** Current sequence element, $\boldsymbol{x}_t$ and previous state of the specialist, $\{\boldsymbol{h}_{t-1,k}\}$, for $k \in \{1, \ldots, n_s\}$ and structure of memory as a matrix $M$ with row wise compartmentalized memories, where $m_i$ refers to the state of slot $i$ (total number of slots is $n_m$).

***Step 1: Process image by position $p$ with fully convolutional net***
- $\boldsymbol{c}_p = [\mathrm{CNN}(\boldsymbol{x}_t)]_p$
- $\boldsymbol{z}_t = [\boldsymbol{c}_p \, \boldsymbol{e}_p]$   *(concatenate encoding of position to CNN output)*

***Step 2: Specialists compete to be selected to update the workspace based on current input***
- $\boldsymbol{q}_k = \boldsymbol{h}_{t-1,k} \boldsymbol{W}^q$
- $s_k = \mathrm{softmax}\left(\frac{\boldsymbol{q}_k \boldsymbol{\kappa}}{\sqrt{d_e}}\right)$, where $\boldsymbol{\kappa} = (\boldsymbol{z}_t \boldsymbol{W}^e)^{\mathrm{T}}$
- Construct a set $\mathcal{F}_t$ which contains the indices of the $n_{\mathrm{sel}}$ specialists that have the largest $s_k$
- $\bar{\boldsymbol{h}}_{t,k} = \begin{cases} g_k\left(s_k \boldsymbol{z}_t \boldsymbol{W}^v, \boldsymbol{h}_{t-1,k}\right) & k \in \mathcal{F}_t\,, \\ \boldsymbol{h}_{t-1,k} & k \notin \mathcal{F}_t\,, \end{cases}$
- $\boldsymbol{a}_k = s_k \boldsymbol{z}_t \boldsymbol{W}^v \; \forall\, k \in \mathcal{F}_t$ (Scaled Dot Product Attention)

***Step 3: Activated specialists write in a shared workspace***
- $\widetilde{\boldsymbol{Q}} = \boldsymbol{M} \widetilde{\boldsymbol{W}}^q$
- $\boldsymbol{R} = [\boldsymbol{M}; \boldsymbol{A}]$ where $\boldsymbol{A}$ is the matrix whose rows are the $\boldsymbol{a}_k \forall k \in \mathcal{F}_t$
- $\boldsymbol{M} \leftarrow \mathrm{softmax}\left(\frac{\widetilde{\boldsymbol{Q}}(\boldsymbol{R}\widetilde{\boldsymbol{W}}^e)^{\mathrm{T}}}{\sqrt{d_e}}\right) \boldsymbol{R} \widetilde{\boldsymbol{W}}^v$

***Step 4: Broadcast of information from the shared workspace***
- $\widehat{\boldsymbol{q}}_k = \bar{\boldsymbol{h}}_{t,k} \widehat{\boldsymbol{W}}^q \quad \forall k \in \{1, \ldots, n_s\}$
- $s_{k,j} = \mathrm{softmax}\left(\frac{\widehat{\boldsymbol{q}}_k \widehat{\boldsymbol{\kappa}}_j}{\sqrt{d_e}}\right)$ where $\widehat{\boldsymbol{\kappa}}_j = (\boldsymbol{m}_j \widehat{\boldsymbol{W}}^e)^{\mathrm{T}} \quad \forall k \in \{1, \ldots, n_s\},\, j \in \{1, \ldots, n_m\}$
- $\boldsymbol{h}_{t,k} = \bar{\boldsymbol{h}}_{t,k} + \sum_j s_{k,j} \widehat{\boldsymbol{v}}_j$ where $\widehat{\boldsymbol{v}}_j = \boldsymbol{m}_j \widehat{\boldsymbol{W}}^v \quad \forall k \in \{1, \ldots, n_s\}$

---

## B   HYPERPARAMETERS

Table 3 lists the different hyper-parameters.

**Parameters in RIMs+SW:**

RIMs with shared workspace has three set of parameters:

- **Parameters corresponding to Input attention** Parameters for the attention for the $k$-th specialist $\theta_k = (W_k^q, W^e, W^v)$ corresponding to query, keys, and values respectively. Each specialist has different query parameters but share the same keys and values (which are function of the input). In the table it corresponds to the inp keys, inp values, inp heads respectively.
- **Writing in a shared workspace:** Parameters corresponding to the writing in the memory. Here, we follow the similar mechanisms as in RMC(Santoro et al., 2018), where shared

---

**Algorithm 2:** Shared Workspace integration with TIMs

---

***Notation:*** Consider $\boldsymbol{h}_l$ as the output of the $l^{th}$ transformer layer. Let sequence length of original input be $T$ and embedding dimension of transformer be $D$. Let the transformer be composed of $n_b$ mechanisms and memory be denoted as a matrix $M$ with row wise compartmentalized memories, where $m_i$ refers to the state of slot $i$ (total number of slots is $n_m$). Consider $\boldsymbol{h}_l^k = \boldsymbol{h}_l[:, (k-1)D/n_b : kD/n_b]$ to be the hidden state of mechanism indexed $k$ at layer $l$.

***Initialization:*** Convert the raw input $X \in \mathbb{R}^{T \times vocab\_size}$ to
$\boldsymbol{h}_0 = positional\_encoding + Embedding(X)$ where $\boldsymbol{h}_0 \in \mathbb{R}^{T \times D}$. Initialize memory matrix $M$ which remains common for all layers in the transformer.

***Input to the layer*** $l$***:*** $\boldsymbol{h}_{l-1}$ having shape $\mathbb{R}^{T \times D}$

***Step 1: Mechanisms compete to be selected to update the workspace based on the input they receive from the previous layer***
- $W^c \in \mathbb{R}^{D/n_b \times 1}$
- $c_k = \boldsymbol{h}_{l-1}^k W_k^c \quad \forall k \in \{1, \ldots, n_b\}$
- $c = softmax(concat(c_1, .., c_{n_b})), c \in \mathbb{R}^{T \times n_b}$
- For each time step $t$ in the original sequence of length $T$, we use the soft score $c$ to select the top $n_{sel}$ mechanisms which would self-attend and write to the memory. Hence generating set $\mathcal{F}_t$ which stores the indices of $n_{sel}$ mechanisms for position $t \in \{1, 2, ..., T\}$. Also construct $c_k^* \in \mathbb{R}^{T \times D/n_b}$ where

$$c_k^*[t,:] = \begin{cases} c[t][k] & k \in \mathcal{F}_t \,, \\ 0 & k \notin \mathcal{F}_t \,, \end{cases}$$

***Step 2: Selected mechanisms self-attend and update their hidden state***
- $residual_k = \boldsymbol{h}_{l-1}^k$
- $\bar{\boldsymbol{h}}_l^k = c_k^* \odot SelfAttention(\boldsymbol{h}_{l-1}^k) + residual_k \quad \forall k \in \{1, \ldots, n_b\}$

***Step 3: Selected mechanisms write on the shared workspace***
- Memory matrix $M$ was last modified by mechanisms of layer $l-1$
- Let $\boldsymbol{a}_k = c_k^* \odot \bar{\boldsymbol{h}}_l^k$ and $\boldsymbol{a} = concat(\boldsymbol{a}_1, .., \boldsymbol{a}_{n_b})$. Absorb the first dimension (corresponding to position in the sequence) in the batch dimension by reshaping $\boldsymbol{a}$. Perform the same steps as in algorithm 1.
- $\widetilde{\boldsymbol{Q}} = M\widetilde{\boldsymbol{W}}^q$
- $\boldsymbol{R} = [M; A]$ where $\boldsymbol{A} = \boldsymbol{a}W^v$
- $M \leftarrow softmax\left(\frac{\tilde{\boldsymbol{Q}}(\boldsymbol{R}\widetilde{\boldsymbol{W}}^e)^{\mathrm{T}}}{\sqrt{d_e}}\right) \boldsymbol{R}\widetilde{\boldsymbol{W}}^v$

***Step 4: Broadcast of information from the shared workspace***
- Reshape the new memory to bring back the sequence dimension. Perform the same steps as in algorithm 1.
- $\hat{\boldsymbol{q}}_k = \bar{\boldsymbol{h}}_l^k \widehat{\boldsymbol{W}}^q \quad \forall k \in \{1, \ldots, n_b\}$
- $s_{k,j} = softmax\left(\frac{\hat{\boldsymbol{q}}_k \hat{\boldsymbol{\kappa}}_j}{\sqrt{d_e}}\right)$ where $\hat{\boldsymbol{\kappa}}_j = (\boldsymbol{m}_j \widehat{\boldsymbol{W}}^e)^{\mathrm{T}} \quad \forall k \in \{1, \ldots, n_b\}, \, j \in \{1, \ldots, n_m\}$
- $\boldsymbol{h}_l^k = \bar{\boldsymbol{h}}_l^k + \sum_j s_{k,j} \hat{\boldsymbol{v}}_j$ where $\hat{\boldsymbol{v}}_j = \boldsymbol{m}_j \widehat{\boldsymbol{W}}^v \quad \forall k \in \{1, \ldots, n_b\}$

---

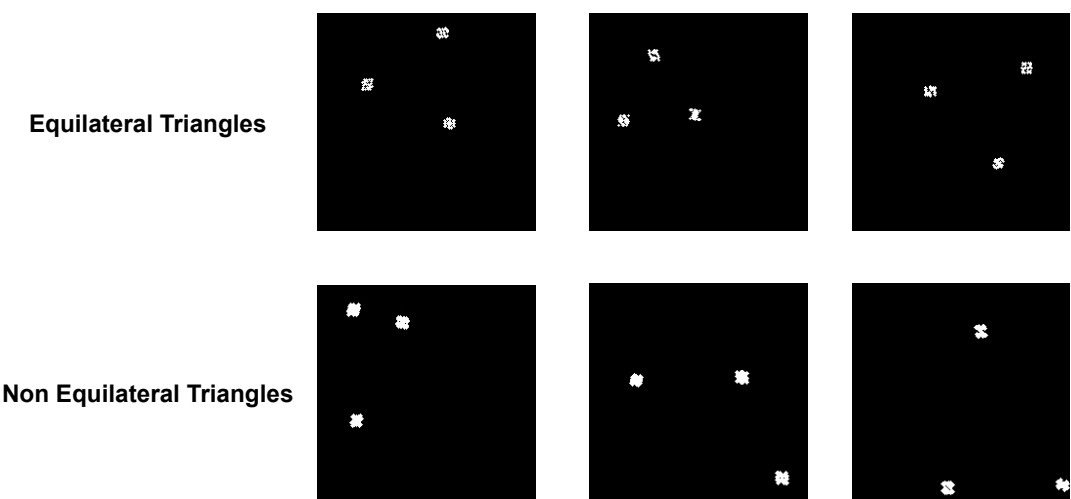

**Equilateral Triangles**

**Non Equilateral Triangles**

Figure 6: A demonstration of the detecting equilateral triangles task.

| Parameter | Value |
| --- | --- |
| Number of specialists ($n_s$) | 6 |
| Size of each specialist | 85 |
| Number of memory slots ($n_m$) | |
| Optimizer | Adam(Kingma and Ba, 2014) |
| learning rate | $1 \cdot 10^{-4}$ |
| batch size | 64 |
| Inp keys | 64 |
| Inp Values | 85 |
| Inp Heads | 4 |
| Inp Dropout | 0.1 |
| Number of memory slots | 4 |
| Number of memory heads | 1 |
| Size of attention head | 32 |
| Key size | 32 |
| Number of MLP layers in Attention | 3 |
| Gate Style | 'unit' |
| Memory Attention Heads | 4 |
| Memory Attention keys | 32 |
| Memory Attention Values | 32 |

Table 3: Generic Hyperparameters for the proposed model (for RIMs)

workspace is seen as a Matrix with row wise compartmentalized memories (i.e slots) i.e $\widetilde{\boldsymbol{W}}^q, \widetilde{\boldsymbol{W}}^e, \widetilde{\boldsymbol{W}}^v$. In the table it corresponds to number of memory slots, number of memory heads, size of attention head, key size and number of mlp layers in attention. These are the same hyper-paramter as in RMC (Santoro et al., 2018). We tried two different set of hyper-parameters (a) where we only have a single slot and (b) where we have 4 slots.

- **Broadcast of Information from the shared workspace**: In this process, the information in the workspace gets broadcasted to all the specialists such that each specialist produces a query, and the keys and values are a function of the memory state. Each specialist gets information from the memory according to its query, and this information is used to update the state of each specialist in a residual fashion. This corresponds to the parameters of $\widehat{\boldsymbol{W}}^v$, $\widehat{\boldsymbol{W}}^q, \widehat{\boldsymbol{W}}^e$ in the table i.e memory attention heads, memory attention keys, and memory attention values. We did not do any hyper-parameter search for these hyper-parameters.

**Resources Used:**

- For vision tasks like Sort-of-clever, Equilateral triangle, CIFAR classification, it takes about 6 hours to run 200 epochs on V100 (32G) GPU.

- It takes about 2 days to train the proposed model on bouncing ball task for 100 epochs on V100 (32G) GPU. We did not do any hyper-parameter search specific to a particular dataset (i.e 4Balls or 678Balls or Curtain Task). We ran the proposed model for different number of memory slots (i.e 2/4/8) for all the different datasets.

- For Starcraft task, it takes about 5 days to train on V100 (16G) GPU with batch size of 4.

## C   IMPLEMENTATION DETAILS

**Writing Information in the shared workspace**. While writing information to the shared workspace, we update the workspace using a gating mechanism as proposed in Santoro et al. (2018). The gating mechanism consists of *input* and *forget* gates. Let $M^{t-1}$ and $M^t$ be the previous and updated memory matrix respectively. Let $M$ be the result of the attention mechanism as described in step 2 of section 2.1. Let $X_{1...n_s}$ be the input to $n_s$ specialists. The gating mechanism can be formulated as follows.

$$\bar{X} = \frac{1}{n_s} \sum_{i=1}^{n_s} \text{relu}(X_i \times W^1)$$
$$K = \bar{X} + \tanh(M^{t-1})$$
$$I = \text{sigmoid}(KW^I)$$
$$F = \text{sigmoid}(KW^F)$$
$$M^t = I \times \tanh(M) + F \times M^{t-1}$$

Here, $I$ and $F$ indicate the input and forget gates respectively. Note that $W^1$ is shared across all $n_s$ specialists.

## D   PROPERTIES OF SHARED WORKSPACE

In section 2, we claim that higher-order interaction terms and effects due to persistence of memory are key contributors to Shared Workspace performance. We support those claims here:

**Shared Workspace vs repeated self attention** Higher-order interaction can be simulated by repeating the self-attention step multiple times at the same layer/time-step. However, due to the absence of a global communication channel, there is no constraint that the messages passed among the neural modules should lie in the same representation space. We modify a standard transformer where we repeat the self-attention step two times in every layer. We expect that 2×Self Attention will perform worse than SW. We also run a model where both self-attention as well as shared workspace is used by the transformer to update its state.

**Persistence of Memory** To check whether persistence is crucial for our model to perform well, we run a model where we re-initialize the shared workspace at every layer. Again we expect that removing memory persistence should result in a drop in performance and speed of convergence.

We run these models on sort-of-clevr dataset and present the results in figure 7

We note that removing persistence of memory results in significantly slower convergence. Replacing SW with 2×SA results in a significant drop in performance.

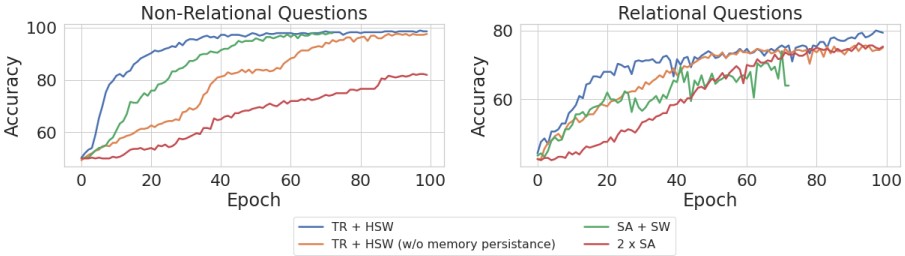

Figure 7: **Comparison on Sort-of-CLEVR relational reasoning**. Speed of convergence for relational and non-relational questions in the sort-of-clevr dataset. We can see that the Shared Workspace model converges faster and generalizes better as compared to all the other models. Here SW refers to shared workspace, $2\times$SA refers to applying self-attention twice in the same layer, SW+SA refers using both Shared Workspace and Self Attention in each transformer layer.

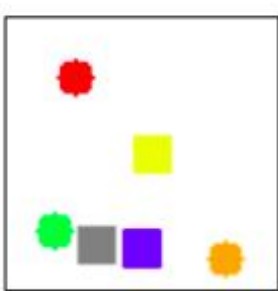

**Relational questions:**

1. What is the shape of the object closest to the red object? ⇒ square
2. What is the shape of the object furthest to the orange object? ⇒ circle
3. How many objects have same shape with the blue object? ⇒ 3

**Non-relational questions:**

1. What is the shape of the red object? ⇒ Circle
2. Is green object placed on the left side of the image? ⇒ yes
3. Is orange object placed on the upside of the image? ⇒ no

Figure 8: A sample from the sort-of-clevr dataset.

# E  TRANSFORMER TASKS

## E.1  DETECTING EQUILATERAL TRIANGLES

A demonstration of this task can be found in figure 6. We use images of size $64 \times 64$ for this task. Our training dataset consists of 50000 examples and we evaluate on 10000 examples. We follow the same setup as vision transformers Dosovitskiy et al. (2020) for this task. We divide the image into patches of size $4 \times 4$, this sequence of patches is fed as input to a 4-layered transformer along with the CLS token which is used for classification. We set hidden dim to 256 and ffn dim to 512. For the proposed model (TR+SSW, TR+HSW), We use a query and key size of 32, and value size of 64. We use 4 heads during reading from and writing into the shared workspace which consist of 8 memory slots. For the baseline models (TR, TR + HC, STR), we use query, key and value size of 64 and 4 heads. For training, we use a batch size of 64. We train the model for 200 epochs using Adam optimizer with a learning rate of 0.0001. We anneal the learning rate using cosine annealing.

## E.2  SORT-OF-CLEVR

Figure 8 shows a sample from this dataset. The images in this dataset are of size $75 \times 75$. Each question is encoded into 11 bits. The first 6 bits indicate color, the next 2 bits indicate question type (relational or non-relational), and the remaining 3 bits indicate question subtype (according to figure 8). We use a 4-layered transformer for this task with hidden dim set to 256 and ffn dim set to 512. For the proposed model (TR+SSW, TR+HSW), We use a query and key size of 32, and value size of 64. We use 4 heads during reading from and writing into the shared workspace which consists of 8 memory slots. For the baseline models (TR, TR + HC, STR), we use query, key and value size of 64 and 4 heads. We encode the 11 bit question into a 256 dimensional vector representation and concatenate it with the sequence of $15 \times 15$ sized patched obtained from the image.

We use the representation corresponding to the CLS token for classification. We train the model using cross-entropy loss. We use a batch size of 64 and train the model for 100 epochs. We use Adam optimizer with a learning rate of 0.0001 for training.

### E.3 CATER: OBJECT TRACKING

Each CATER video consists of about 300 frames of size $224 \times 224$. We first sample frames at a sampling rate of 6 which results in 50 frames. From these 50 frames, we stack 5 consecutive frames together and pass each stack through a 18 layered resnet. The corresponding sequence of 10 frames is passed as input to the transformer. This task is setup as a classification task where we have to predict which cell in the $6 \times 6$ grid contains the snitch in the final frame. We use a 6-layered transformer with hidden dim set to 512 and ffn dim set to 2048. For the proposed model (TR+SSW, TR+HSW), We use a query and key size of 32, and value size of 64. We use 8 heads during reading from and writing into the shared workspace which consists of 8 memory slots. For the baseline models (TR, TR + HC, STR), we use query, key and value size of 64 and 8 heads.

## F RIMs TASKS

### F.1 BOUNCING BALL

The dataset consists of 50,000 training examples and 10,000 test examples showing ∼50 frames of either 4 solid balls bouncing in a confined square geometry (*4Balls*), 6-8 balls bouncing in a confined geometry (*678Balls*), 3 balls bouncing in a confined geometry with an occluded region (*Curtain*), or balls of different colors (*Colored 4Balls*) and (*Colored 678Balls*). We train baselines as well as the proposed shared workspace extension (e.g., RIMs + SW). As shown in Fig. 9, we study the performance of the proposed model compared with LSTM, RIMs and RMC. The first 10 frames of ground truth are fed in and then the system is rolled out for the next 35 time steps. During the rollout phase, the proposed method performs better than the baselines in accurately predicting the dynamics of the balls as reflected by cross entropy (CE).

We trained baselines as well as proposed model for about 100 epochs. We use the same architecture for encoder as well as decoder as in (Van Steenkiste et al., 2018). Hyper-parameters specific to the proposed architecture are listed in Tab. 3.

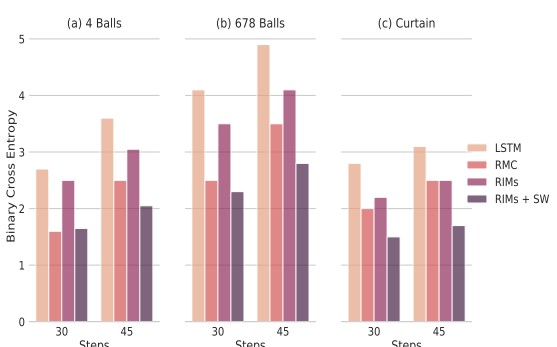

Figure 9: **Bouncing ball motion:** Prediction error comparison of the proposed method, LSTM, RIMs and RMC baseline. Given 10 frames of ground truth, the model predicts the rollout over the next 35 steps. Here, we present the BCE for the 30th frame and 45th frame. The *proposed SW extension performs better than other baselines in accurately predicting the dynamics, with an increasing advantage as the number of unrolled steps (30 vs 45) and balls ((a) vs (b)) increases*. Results are an average over 5 random seeds.

## G INTEGRATING SW WITH MORE ARCHITECTURES

### G.1 TIMs

TIMs was proposed by Lamb et al. (2021). A transformer network is divided into 'independent mechanisms' which update their state via sharing information between positions and sharing information between mechanisms. The information sharing step between mechanisms can be replaced by SW to create TIMs+SW.

### G.1.1 MULTIMNIST GENERATION

In this task, we train an Image Transformer Parmar et al. (2018) (pixel-by-pixel, raster-order generative model) for next pixel prediction task on the "MultiMNIST dataset"

| Parameter | Value |
|---|---|
| **Common Parameters** | |
| Optimizer | Adam(Kingma and Ba, 2014) |
| Learning rate | $1 \cdot 10^{-3}$ |
| Batch size | 12 |
| Number of attention heads | 8 |
| **TR** | |
| Size of transformer layer | 256 |
| **TIMs** | |
| Number of mechanisms | 4 |
| Size of mechanism | 48 |
| **TIMs+SW** | |
| Number of mechanisms | 4 |
| Size of mechanism | 40 |
| Number of memory slots | 2 |
| Size of memory slots | 160 |
| Memory Attention Heads | 8 |
| Gate Style | 'unit' |
| Number of MLP layers in Attention | 2 |

Table 4: Hyperparameters for MultiMNIST Task

Each $32 \times 32$ image in this dataset is made up of four randomly selected (and augmented) MNIST digits (resized to $32 \times 8$) placed side-by-side as shown in figure 10. The digits themselves are selected independently of one-another.

The main aim of creating such a task is to observe the working of independent mechanisms in architectures such as TIMs (Lamb et al., 2021). Each image in the MultiMNIST dataset can be broken down into different sets of independent spatial components. Since the digits which make up the image are independently selected, the joint distribution of pixel intensities in any one of the four sections of the image is statistically independent of the pixel intensities in any other section of the image. Moreover each section of the image can be further broken down into independent spatial components: one that pertains to the background and one that pertains to the foreground.

It is expected that a monolithic architecture (having a single computational unit) would have to devote a significant portion of its training to learn the statistical independence between the different constituents of the image. On the other hand, architectures made up of sparsely interacting independent mechanisms have a natural way of capturing such statistical independence. A division of labour where each mechanism is focused on the generation of a distinct independent constituent of the image should allow for better generalization on the test set. Once the generation of a constituent is completed, the task can be handed over to some other mechanism based on current position in the image.

For this experiment we train a standard transformer with shared parameters across all layers (denoted by TR), TIMs (Lamb et al., 2021) with 4 mechanisms, and a modified version of TIMs with 4 mechanisms where the pair-wise communication between the mechanisms is replaced by communication via a shared workspace (denoted by TIMs+SW).

**Training.** We follow the minGPT Image Transformer setup Karpathy (2020) for our experiments. All three of the configurations have 8 layers, 8 heads for multi-headed attention and use the exact same parameter initialization and base architecture. We train all three of the models for 20 epochs.

In the TR model, all of the 8 monolithic layers share the same set of parameters. In TIMs and TIMs+SW, the first two layers are the standard monolithic layers having shared parameters. The middle four layers in both of these architectures are modular layers with four mechanisms. These four

| Model | Loss |
|---|---|
| TR | 0.000058 |
| TIMs (4 mechanisms) | 0.000050 |
| TIMs+SW (4 mechanisms) | 0.000042 |

Table 5: **MultiMNIST Generation Task:** We report cross-entropy loss between the generated pixel values and the true pixel values on the test set of MultiMNIST Generation Task (smaller numbers are better)

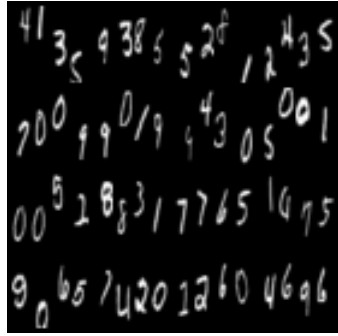

Figure 10: A randomly selected batch of 16 images from the MultiMNIST generation dataset (4 rows and 4 columns)

layers share the same set of parameters. In the case of TIMs+SW, the four mechanisms in these layers communicate via a shared workspace (having 2 memory slots). This shared workspace is common for all four middles layers and is absent in TIMs where the mechanisms communicate via pair-wise competition as proposed in the original paper. TIMs and TIMs+SW architectures are concluded by two more monolithic layers which again share the same parameters.

For all three models to have comparable number of parameters, we chose the transformer embedding dimension to be 256 for TR model, 192 for TIMs model and 160 for TIMs+SW model. In TIMs and TIMs+SW, the embedding dimension is divided equally among the four specialists. Each memory slot in the shared workspace of the TIMs+SW model has a 160 dimensional embedding and the model uses four heads to perform read and write operations on the shared workspace. Total number of parameters for all three architectures lie between 1M and 1.8M.

**Results.** We observe the best cross-entropy loss in 20 epochs on the test set of the MultiMNIST dataset for the next pixel prediction task in the table 5. We further plot the sixth layer "mechanism activation score" of TIMs and TIMs+SW while generating the first four images of the test set in the best epoch (shown in figure 5).

### G.1.2 USING WORKSPACE FOR LANGUAGE MODELLING

We train our models on the WikiText-103 dataset by posing a language modeling problem. The dataset is divided into train, test and validation sets which are composed out of 28,475, 60 and 60 articles respectively. The total number of tokens in the train set is more than 103 million, hence the name of the dataset. This dataset retains numbers, punctuation and case.

**Training.** We train our models for 15 epochs for the next word prediction task on the WikiText-103 dataset and report the perplexity on the validation set. We show the results using TIMs (Lamb et al., 2021) with 4 mechanisms and TIMs+SW with 4 mechanisms (where we replace the pairwise communication in TIMs with communication via a shared workspace like in the MultiMNIST experiment). We modify the FAIRSEQ Ott et al. (2019) transformer language model class for all of our experiments.

For TIMs+SW, we train and test two different variants: TIMs+SSW uses soft attention to generate the activation scores of competing independent mechanisms whereas TIMs+HSW uses top-k attention with k=2.

Since in this test, our aim is to compare the performance of the two models for the language modeling task, the architectures are only made up of a transformer decoder. In both of the models, there are 8 transformer decoder layers divided into 3 sets. The first 2 layers are standard monolithic decoder layers which share the same parameters. The next 4 layers are modular layers (TIMs layers or TIMs+SW layers depending on the model choice). These layers also share the same parameters among themselves. The last 2 layers are again standard monolothic decoder layers, both sharing the same parameters.

The inputs to the network are $1024$ dimensional word embeddings, input to a transformer layer of dimension $1024$ and feed forward dimension of $2048$.

Both of the networks have 8 attention heads with head dimension of 128. The total transformer layer size of $8 \times 128 = 1024$ is equally divided among the four mechanisms. In the case of TIMs, these mechanisms (in layers 3,4,5) interact via pair-wise communication, whereas in TIMs+SSW and TIMs+HSW, these mechanisms interact via a shared workspace. The shared workspace has 2 memory slots, each 1024 dimensional, having 4 attention heads for reading and writing.

| Parameter | Value |
|---|---|
| **Common Parameters** | |
| Optimizer | Adam(Kingma and Ba, 2014) |
| Learning rate | $5 \cdot 10^{-4}$ |
| Adam betas | $0.99, 0.98$ |
| Weight decay | $0.01$ |
| lr scheduler | 'inverse square root' |
| Max tokens per gpu | 3078 |
| Batch size multiple | 8 |
| Number of attention heads | 8 |
| Transformer layer size | 1024 |
| Number of Mechanisms | 4 |
| Update frequency | 4 |
| Number of warmup updates | 4000 |
| Starting Warmup lr | $1 \cdot 10^{-7}$ |
| **TIMs+SSW** | |
| Number of memory slots | 2 |
| Size of memory slots | 1024 |
| Memory Attention Heads | 4 |
| Gate Style | 'unit' |
| Number of MLP layers in Attention | 3 |
| top-k competition | False |
| **TIMs+HSW** | |
| Number of memory slots | 2 |
| Size of memory slots | 1024 |
| Memory Attention Heads | 4 |
| Gate Style | 'unit' |
| Number of MLP layers in Attention | 3 |
| top-k competition | True, k=2 |

Table 6: Hyperparameters for WikiText-103 Language Modeling Task

**Results.** We plot the perplexity (per epoch) on the validation set. All models have comparable number of parameters (within a 10% difference). We note that TIMs performs poorly on this dataset but adding shared workspace improves the performance consistently. We also note that sparsity indeed helps as TIMs+HSW performed the best.

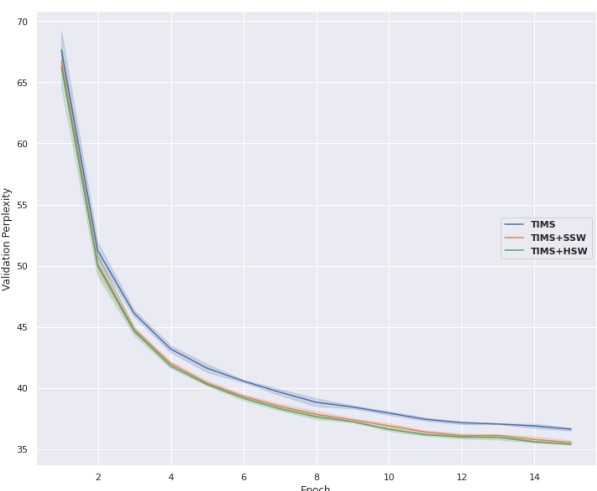

Figure 11: Per epoch validation perplexity for TIMs, TIMs+SSW, TIMs+HSW for wikitext-103 language modeling task

