# OpenReview forum: "Coordination Among Neural Modules Through a Shared Global Workspace"
_ICLR.cc/2022/Conference — ICLR 2022 Oral_

### Official Review · Reviewer_Cdd9 · 2021-11-01

**Correctness:** 4
**Technical Novelty And Significance:** 4
**Empirical Novelty And Significance:** 4
**Recommendation:** 10
**Confidence:** 3

**Details Of Ethics Concerns:**

No concerns.

**Main Review:**

Strengths
1. Novel and significant contribution by proposing a new mechanism for inter-subnets information exchange/learning.
2. Of high relevance to the ICLR community. The generic approach cuts across different AI discipline as the experiments are performed on vision, language and reasoning tasks.
3. Experiments are comprehensive and well-explained.

Weaknesses
Minor: Some technical details are not elaborated for readers who are less familiar with the respective topics. For example, the section on the key-query-value attention mechanism is quite brief. It's the key contribution on the success of the proposed approach.

**Summary Of The Paper:**

Paper proposes a novel mechanism for information exchange between different neural subnets. It replaces the pairwise interactions with a share memory space. The memory is updated by a competitive scheme which the top-k updates are selected via the key-query-value
attention mechanism. The shared memory/workspace then broadcast the updated state to all other upstream specialist subnets. Each specialist subnet then update its representation/weights accordingly.

Paper claims 3 advantages of this approach:
1. Higher-order (HO) interaction among neural modules
2. Dynamic filtering due to persistence of memory.
3. Lower computational complexity of using shared workspace for synchronizing different specialists

Experimental results are very promising across a wide variety of tasks, namely DETECTING EQUILATERAL TRIANGLES; SORT-OF-CLEVR; CATER: OBJECT TRACKING;  for transformer architectures. Also, experiments on RIM for the BOUNCING BALL task and TIMS for the MULTIMNIST GENERATION and Language Modeling Task were also done. Finally, the approach was also experimented on model  free RL on ATARI game task.

**Summary Of The Review:**

Paper proposes a novel approach which produces commendable experimental results across a wide range of tasks in vision, language and reasoning. The impact of this work is very significant and is of high relevance to the community. While I am not entirely familiar with the cited prior work on key-query-value attention mechanism which forms the backbone of the proposed approach, the entire paper is very well-written and comprehensible.

---

> ### Author Response · Authors · 2021-11-18
> **Thanks :)**
>
> We thank the reviewer for the review. We are enthused that the reviewer found the experimental results very promising, and is enthusiastic about the paper. We will clarify the technical details in the next version of the paper.

---

### Official Review · Reviewer_2GPN · 2021-11-04

**Correctness:** 4
**Technical Novelty And Significance:** 4
**Empirical Novelty And Significance:** 3
**Recommendation:** 8
**Confidence:** 3

**Main Review:**

Strengths:

- The paper is written very well. The introduction provides a good motivation and challenges and illustrates the use cases clearly using examples. The related work compares the proposed approach with other memory-based neural models. The description of methods is concise, but sufficiently detailed.

- The proposed approach is original to the best of my knowledge. The key idea is very intuitive and motivated by insights from cognitive science literature.

- The authors perform experiments on a wide variety of tasks including a toy task for detecting equilateral triangles, multi MNIST Generation, object tracking, relational reasoning, physical reasoning and Atari video games. I also like that the authors use different types of backbones (Transformers and RIMs) in different experiments which indicates the proposed Shared Workspace method is not specific to certain kind of backbones.

Weaknesses:

- Although the authors perform experiments with wide variety of tasks, multi-agent tasks are missing where I believe coordination is more important. In all the tasks used in the paper, different neural modules process different parts of the input to make a common prediction. Multi-agent tasks are more challenging as each agent would get a different input but also predict a different action.

- It is unclear how certain hyper parameters are chosen. For example why is the patch size 4x4 in equilateral triangles, 6x6 in object tracking? The number of memory slots, size of memory slots, etc. different across different experiments. How are these chosen?

- The authors propose two versions of Shared Workspaces, soft and hard (SSW and HSW). Some experiments one contain a single version. Why are both SSW and HSW not evaluated in all the experiments?

**Summary Of The Paper:**

This paper presents a method of using external memory called "shared workspace" for communication among different neural modules or "specialists". The key idea is that there are limits on the communication bandwidth and the specialist modules must compete for access. The communication limit encourages specialization and compositionally and facilitate better synchronization. Experiments over a variety of tasks indicate the proposed Shared Workspace model improves performance over baselines.

**Summary Of The Review:**

The paper proposes a novel method for coordination between neural modules which is well motivated. The experiments are comprehensive although some details are missing. Addition of multi-agent tasks would make the paper stronger.

Edit after author response:
I read the author response and I would like to maintain my positive rating.

---

> ### Author Response · Authors · 2021-11-18
> **Multi-agent + Hyperparameters + Experiments on BabyAI**
>
> We thank the reviewer for their time in reviewing the paper and providing constructive feedback.
>
> **Multi-agent tasks are more challenging as each agent would get a different input but also predict a different action.**
>
> Evaluating the proposed architecture on mutl-agent RL where different agents can follow different objectives would be an interesting area for future work.  For this work, we have shown the utility of the idea by using it across different architectures (Transformers and slot based models), and different tasks.
>
> **Regarding Hyperparameters**
>
> For selecting the hyper-parameters, we tried various different values for each hyper-parameter for the baselines as well as the proposed method. For example, for the patch size we tried 4 x 4, 6 x 6, 8 x 8, 16 x 16 and found that 4 x 4 worked best for detecting equilateral triangles. For object tracking, 6 x 6 is the inherent grid size in the dataset which we do not modify while the image size is 224 x 224. We process each image in cater object tracking using a convolutional network and do not split it into patches before processing.  For the memory slots, we experimented with various values and have reported the best for each experiment. We have also presented an ablation over the number of memory slots for the bouncing balls experiments where we show that 5 slots work best.
>
> **more experiments on BabyAI**
>
> To further show that the proposed model is useful in more general cases we conduct experiments on the babyai framework[1]. BabyAI is a benchmark for studying the sample efficiency for RL algorithms for grounded language learning. Here, an agent is navigating the environment with the aim to complete its goal described by natural language. The tasks in this benchmark are procedurally generated and hence zero-shot generalization becomes a challenge for RL algorithms.
>
> In Babyai, we evaluate the model on different environments than those we used for training. We train on - GoToObj, GoToRedBall, GoToRedBallGrey, GoToLocal, PutNextLocal, PickUpLoc, GoToObjMaze, GoTo, Pickup, UnblockPickup, Open, Unlock, PutNext, Synth, SynthLoc, GoToSeq, SynthSeq, GoToImpUnlock. We evaluate on - BossLevel. We also evaluate a harder version on BossLevel (which we call BossLevelGen) where we have more rooms and larger room sizes. Thus this is a zero-shot generalization setting. The goal behind testing on multi-task setup is to see if the division of labour among different modules can help the policy to generalize in a zero-shot manner to more complex tasks (BossLevel).
>
> | Model | Num Modules | BossLevel | BossLevelGen |
> | -------- | ------------------- | -------------  | ------------------- |
> | RIM          |    10           |      0.28     |   0.09               |
> | RIM + SW |  10            |      0.35     |    0.21              |
>
> We can see that RIM with shared workspace convincingly outperforms RIMs with pairwise communication. Also, babyai is a general navigation environment that is not sparse thus confirming that shared workspace does show promising improvements in more general settings and offers superior generalization performance.
>
> We would like to point out that we do provide results on language modeling in appendix section G.1.2 where we show that replacing the pairwise communication in TIMs with shared workspace style attention results in a lower perplexity. We would also like to point out that both atari and bouncing balls are settings where the performance does not depend on small portions of the input. For bouncing balls one needs the ability to model higher-order interactions (since more than 2 balls may collide at the same time) which is not present in pairwise attention but is possible using a shared workspace.
>
> [1] https://arxiv.org/abs/1810.08272

---

### Official Review · Reviewer_pjX4 · 2021-11-05

**Correctness:** 4
**Technical Novelty And Significance:** 4
**Empirical Novelty And Significance:** 3
**Recommendation:** 6
**Confidence:** 4

**Main Review:**

Strengths

1. Interestingness of the approach

The proposed method is appealing since it applies a theory from cognitive science (the global workspace theory) to information processing in networks. The implementation of the idea is quite simple - individual ‘specialist’ units interact with a shared memory layer instead of with each other directly, and the shared memory then broadcasts information back to each of the specialists. The exact means by which this interaction is carried out is by using key-value attention. This approach when applied to transformers leads to much higher computational efficiency (linear instead of quadratic in the sequence length).
Further research in probing similar ideas for much more efficient information processing, backed by ideas from cognitive science, would be very beneficial for the field in my view and this paper would help the community in that regard.

2. Experimental Evaluation

The paper includes exhaustive experimental evaluation over 5-6 environments (including object tracking, and relational reasoning) where the solution requires considering a small portion of the input data, the authors demonstrate that adding the shared workspace model to attention based architectures like Transformers and RIMs leads to superior asymptotic performance, and faster learning. These environments include object tracking (CATER), relational reasoning (sort of CLEVR) and Atari games. The authors also include details about the experiments and algorithm implementation in the appendix, which will aid in reproducibility.

Weaknesses

1. Unclear if the approach will scale to larger, more unstructured datasets (where current attention based architectures have already been shown to thrive)

Most of the experiments considered involve problems that only require a small portion of the input to be solved (eg: just the patch with the points for the equilateral triangles, or just the target object for Cater object tracking). I am concerned whether the framework proposed here will also be effective in settings where this is not necessarily the case, such as general modeling of language and images. For example, adding shared workspace to transformers imposes a communication bottleneck between representations at different positions of the sequence. It is possible that for problems where the solution does not depend on only a small portion of the input, considering the pairwise relationships of representations at every point in the sequence is critical for good performance.
To study this, the shared workspace model would have to be evaluated on larger, more unstructured datasets, where transformer based architectures have already been demonstrated to do well (for example the data on which GPT-2 was trained). Adoption of this approach would be much more widespread if the authors can demonstrate that on these larger datasets, training transformers with the shared workspace doesn’t lead to worse performance than training the regular transformer based models that are currently used.

**Summary Of The Paper:**

The paper proposes a modification for attention-based network architectures drawing inspiration from the global workspace theory in cognitive science. Essentially connections are made sparser, with different ‘specialist’ units communicating with each other through a limited bandwidth channel.


**Summary Of The Review:**

The paper proposes an implementation of an interesting theory from cognitive science for more efficient information processing in networks, by making connections between entities sparser. While there is extensive evaluation on environments, most of these involve processing a small portion of the input. The paper will be made a lot stronger if the approach can be shown to scale to larger more unstructured datasets where transformers are known to work well.

---

> ### Author Response · Authors · 2021-11-18
> **More experiments on BabyAI**
>
> We thank the reviewer for their time in reviewing the paper and providing feedback. We are enthused that the reviewer find the proposed method  appealing. We ran more experiments to address the concerns of the reviewer.
>
> **approach will scale to larger, more unstructured datasets**
>
> To further show that the proposed model is useful in more general cases we conduct experiments on the babyai framework [1]. BabyAI is a benchmark for studying the sample efficiency for RL algorithms for grounded language learning. Here, an agent is navigating the environment with the aim to complete its goal described by natural language. The tasks in this benchmark are procedurally generated and hence zero-shot generalization becomes a challenge for RL algorithms.
>
> In Babyai, we evaluate the model on different environments than those we used for training. We train on - GoToObj, GoToRedBall, GoToRedBallGrey, GoToLocal, PutNextLocal, PickUpLoc, GoToObjMaze, GoTo, Pickup, UnblockPickup, Open, Unlock, PutNext, Synth, SynthLoc, GoToSeq, SynthSeq, GoToImpUnlock. We evaluate on - BossLevel. We also evaluate a harder version on BossLevel (which we call BossLevelGen) where we have more rooms and larger room sizes. Thus this is a zero-shot generalization setting. The goal behind testing on multi-task setup is to see if the division of labour among different modules can help the policy to generalize in a zero-shot manner to more complex tasks (BossLevel).
>
> | Model | Num Modules | BossLevel | BossLevelGen |
> | -------- | ------------------- | -------------  | ------------------- |
> | RIM          |    10           |      0.28     |   0.09               |
> | RIM + SW |  10            |      0.35     |    0.21              |
>
> We can see that RIM with shared workspace convincingly outperforms RIMs with pairwise communication. Also, babyai is a general navigation environment that is not sparse thus confirming that shared workspace does show promising improvements in more general settings and offers superior generalization performance.
>
> We would like to point out that we do provide results on language modeling in appendix section G.1.2 where we show that replacing the pairwise communication in TIMs with shared workspace style attention results in a lower perplexity. We would also like to point out that both atari and bouncing balls are settings where the performance does not depend on small portions of the input. For bouncing balls one needs the ability to model higher-order interactions (since more than 2 balls may collide at the same time) which is not present in pairwise attention but is possible using a shared workspace.
>
> [1] https://arxiv.org/abs/1810.08272

---

### Official Review · Reviewer_EPxc · 2021-11-05

**Correctness:** 2
**Technical Novelty And Significance:** 3
**Empirical Novelty And Significance:** 3
**Recommendation:** 6
**Confidence:** 3

**Main Review:**

I enjoyed reading this paper! I think the strengths fall into three main categories: good problem setup that took the reader from concept to formulation well, an interesting idea with good scoping, and an impressive experimentation suite. More details:

**Good problem setup:** I found the discussion of advantages of a shared workspace compelling. The proposed structure is intuitive and simple, but nonetheless convincing about its claimed advantages of higher-order communications, dynamic filtering, and complexity advantages. The qualitative parts of the writing are strong, effectively building a case for reading this paper.

**Interesting idea with good scoping:** I appreciate that this is one architectural change that considers all inputs and outputs, and shows its advantages over the right baselines. This work avoids the trap of making minor changes and then being unable to really ablate them. The paper does seem to be a unification of existing ideas, but I distinguish that from (and prefer it over) merely concatenating existing ideas.

**Impressive experimentation suite:** There are of course many experiments with thematic consistency, which is great - always better than papers with two testbeds and virtually the same experiment over and over. The first four testbeds make a nuanced point about sparse inputs, and the results on Atari are particularly exciting because they make a general claim and show significant improvement.

My critiques are as follows, in order of significance:

**Content**
1. Main critique: the goals of this paper seem to be to show that accuracy and performance improve **overall**, with a mechanism of higher order communications, dynamic filtering, and linear complexity that we will be convinced are attributes of the SW. This paper doesn't quite get there, though it gets part of the way there.

From my understanding, Triangles, CATER, and Sort-of-CLEVR all tell us that the higher-order communication and single channel will help identify relevant information earlier in the pairwise communication, leading to faster convergence - *if* information is sparse in the input. The MNIST generation experiment tells us something similar about independent regions in inputs. Figs 4 and 5 are convincing to this end. However, the main claim - higher-order communication and a sparser connection of graphs - help for inputs with sparse information in various locations - is less surprising than a claim of general improvement. Even if that's still interesting (it is), the intro and conclusion seem to make a more general claim. Physical reasoning and Atari show more general improvement (and the results on Atari are impressive), but they're also the weakest, most qualitative parts of Sec 4. The ideal solution would be more robust testing of general improvement in these two testbeds, but if not then I would at least appreciate a claim that acknowledges the common and specific nature of the first four testbeds.

This is the crux of the motivation behind my score: I worry that this paper overstates its claim and the real claim is elegant but not that surprising. It's made even murkier by a couple promising results that receive little attention.

2. Relatedly, the Atari section makes a claim about considerable improvement due specifically to more appropriate modularization. That would be cool and a real testament to this approach, but I would need more specific experimentation to demonstrate that the effect really was more appropriate modularization.

**Presentation**
1. The mathematical formulation is quite clear, but it's not presented clearly. I had to puzzle over the paragraph for a while, not because it was overly complicated but because it took a lot of treasure hunting to find all the parts and put them together. The issue is that the *Notation* and *Step 1/2/3* sections write the math into large blocks of text without pause. It would be more helpful to present some end-to-end equations, then break those down in text. It can be unclear without doing it myself whether something is being element-wise updated, transformed, dotted, etc.

2. I really appreciate the slew of testbeds. However, that's how they come across: a laundry list of experiments, one after another, even though they support similar claims. That robustness can be shown by having a claims-driven structure for the results section rather than a testbed-driven structure, which to the reader is arbitrary and doesn't deliver the important information as well as it could. E.g. section titles like "Performance advantage on sparse data", then talk about the different results that show it. If one experiment shows multiple advantages, multiple sections can still point to the same figures.

3. The figures are confusing - they are cramped, small, and have lots of acronyms I have to hunt down in various parts of the paper. They're all helpful content-wise, but this could be shown better with something as simple as better aesthetics and tagging.

4. Nit: the text seems to switch between "specialist" and "neural module". Are these the same thing? Is "specialist" an abstraction/metaphor for "neural module"? Is "neural module" an example of "specialist"? All of those are great, just signpost and explain the switch.




**Summary Of The Paper:**

This paper proposes a communication framework to have multiple modules communicate and switch precedence efficiently, taking inspiration from Global Workspace Theory in cognitive science. The primary contribution is a scheme to replace complete pairwise communications in modularized architectures with a single, limited-capacity workspace that persists and changes over stages of computation. This workspace is implemented with a read/write scheme that iterates through stages of computation (layers in a Transformer or steps in a recurrent architecture): in the write phase, the shared memory is updated according to the current states of the modules that are most informative to the shared memory's current state as determined by a key-value attention scheme, with the modules competing via softmax. In the read phase, the internal states of all modules are updated via another key-value attention scheme. The advantages are claimed to be 1) higher order interaction among modules because every module learns from every other one (at least, more than pairwise), dynamic filtering because the memory persists and updates stage to stage; and linear computational complexity because the number of memory slots doesn't change much (and it's typically small, 1-10).

The related work motivates this paper by the classical AI principle that intelligent systems should have multiple specialized modules rather than one general entity. It distinguishes from prior slot-based memory work in that memory writes here are sparse and competitive, and prior work on reducing computational complexity of Transformer dot product attention through its persistent memory. It aptly assesses itself as a unification of existing ideas.

The experimentation section tests different parts of the proposed scheme. The triangles experiment tests the comparative speed to convergence (and accuracy) of the HO communications here compared to pairwise in baseline Transformer. The MNIST generation experiment with TIMs shows that the shared workspace gives an advantage on domains where input dimensions are mostly independent. The CATER experiment shows a similar result to the triangles experiment (quickly picking out only relevant information) but in a time series, and the Sort-of-CLEVR experiment again reinforces the power of the shared workspace on sparse tasks. The physical reasoning shows general improvement, and the Atari performance shows considerable improvement due to modularization.


**Summary Of The Review:**

**Strengths:**
1. Good problem setup
2. Interesting idea with good scoping
3. Impressive experimentation suite

**Critiques:**
*Content:*
1. The experimentation section mainly convinces me that this architecture is good for accuracy and convergence speed on sparse inputs, not that it's generally advantageous - but the latter seems to be the claim of the paper, and the former is less surprising given the nature of the architecture (though still interesting - it's compelling to see that the architecture works).
2. The experimentation on general improvement is a bit weak.

*Presentation*
1. Formulation is clear, but it's presented piecemeal in long paragraphs that make it hard to parse and put together. Since the formulation is simple (a good, elegant thing), it's frustrating to spend a lot of time hunting down all the pieces.
2. Excellent thoroughness in testbeds, but the results don't have to be presented in such a process-driven way. They can and should be claims/driven.
3. The figures need to be more readable
4. "Specialist"/"neural module" distinction is unclear (see main review)

---

> ### Author Response · Authors · 2021-11-18
> **More experiments on BabyAI + Presentation + Neural Modules/Specialists**
>
> We appreciate that the reviewer enjoyed reading the paper.
>
> The reviewer raises the point that the claims in the paper are not well supported in the paper. Specifically that the proposed model is suited for inputs where there is inherent sparsity and not in more general cases as we claim. We thank the reviewer for allowing us to clarify this. We would first like to explain our thinking behind using the current experiments presented in the paper.  We also ran more experiments on the grid world navigation task, to show that the proposed method achieves better results as compared to using pair-wise interactions.
>
> We would like to point out that shared workspace offers 3 main properties:
>
> - Introduces a communication bottleneck that forces the model to focus on only the most relevant information.
> - Incentivizes division of labor among specialists (modularization)
> - Captures higher-order interactions.
>
> To show the effectiveness of a communication bottleneck, we specifically tested the proposed model on sparse inputs where attending to only relevant information would be necessary. This is the reasoning behind the vision tasks such as cater, sort-of-clevr, etc. To show that the proposed model incentivizes modularization we use the multi-mnist generation task and the bouncing balls task. The ARI measure used in the bouncing balls task measures how well the slots in the base model (in this case the rim modules) are clustered in terms of the objects that they capture. Therefore a higher ARI indicates better clustering, hence better modularization. We can see that using the proposed shared workspace does achieve the best ARI. Capturing higher-order interactions would be particularly useful in environments like atari and bouncing balls since they generally contain interactions of more than 2 objects at a time. We can see that shared workspace communication does have a better performance in both bouncing balls (better ARI [figure 9] and better mse [table 2]) and atari while pairwise communicative attention has a poorer performance which could partly be attributed to its failure to capture higher-order interactions. We would also like to point that bouncing balls and atari are more general frameworks where the information is not sparse. We have also tested the proposed model on language modeling, which is also a more general framework. The results for language modeling are presented in appendix section G.1.2. We can see that using shared workspace in TIMs results in a lower perplexity as compared to pairwise attention.
>
> **Zero shot generalization in BabyAI**
>
> To further show that the proposed model is applicable in more general settings, We use the babyai framework [1]. Here, an agent is navigating the environment with the aim to complete its goal described by natural language. The tasks in this benchmark are procedurally generated and hence zero-shot generalization becomes a challenge for RL algorithms. In babyai, we evaluate the model on different environments than those we used for training. We train on - GoToObj, GoToRedBall, GoToRedBallGrey, GoToLocal, PutNextLocal, PickUpLoc, GoToObjMaze, GoTo, Pickup, UnblockPickup, Open, Unlock, PutNext, Synth, SynthLoc, GoToSeq, SynthSeq, GoToImpUnlock. We evaluate on - BossLevel. We also evaluate a harder version on BossLevel (which we call BossLevelGen) where we have more rooms and larger room sizes. Thus this is a zero-shot generalization setting.The goal behind testing on multi-task setup is to see if the division of labour among different modules can help the policy to generalize in a zero-shot manner to more complex tasks (BossLevel).
>
> | Model | Num Modules | BossLevel | BossLevelGen |
> | -------- | ------------------- | -------------  | ------------------- |
> | RIM          |    10           |      0.28     |   0.09               |
> | RIM + SW |  10            |      0.35     |    0.21              |
>
> We can see that RIMs with shared workspace convincingly outperforms RIMs with pairwise communication (higher is better). Also, babyai is a general navigation environment that is not sparse thus confirming that shared workspace does show promising improvements in more general settings and offers superior generalization performance.
>
> [1] https://arxiv.org/abs/1810.08272
>
> **Regarding Presentation**
>
> We thank the reviewer for giving us thorough feedback regarding the presentation of the paper, we will incorporate the feedback in the next revision of the paper.
>
> Regarding “neural modules” and “specialists” - The reviewer is correct that “neural modules” and “specialists” refer to the same thing. For example, in the bouncing balls experiment each slot that represents a single object is the “neural module” or “specialist”.

---

### Decision · Program_Chairs · 2022-01-20

**Decision:**

Accept (Oral)

**Comment:**

This paper takes inspiration from Global Workspace Theory to propose a modification for attention-based network architectures. This is exemplified both in transformer models and in recurrent models (RIMs). The key idea is to replace the quadratic, pairwise communication between "specialist" units (which in transformers corresponding to the positions) by a higher-order communication model which consists in a competitive, sparse writing step into a shared workspace, followed by a reading step where information is broadcasted from the global workspace to all specialists. The competitive writing step establishes a limited bandwidth channel for this communication which encourages specialization.

The reviewers agree that this is an interesting and very well-written paper which unifies several existing ideas. The main contribution of this paper is in establishing a connection to GWT which may inspire future research to keep developing these ideas. The experiments on relatively small tasks (but challenging ones) provide a good proof of concept. Some concerns pointed out by some of the reviewers include a certain overstatement of the capabilities of the proposed model, as well as lack of experiments that scale up the model to larger and unstructured datasets. The authors replied with additional experiments included in the appendix, which in my opinion address these concerns convincingly.

Overall, this is a strong paper and I recommend acceptance. I encourage the authors to take into account the reviewer's suggestions in the final version. I also think that the connection to related work could be improved, as there is several related works [1, 2, 3] which asks/investigates similar questions to this paper and should probably be acknowledged:
- The "shared global workspace" of this paper (Transformer + SW) is reminiscent of the Star-Transformer [1], as well as other more recent works which use special units (e.g. CLS tokens) to encode "global" representations. While that work does not include the competitive component (the "bottleneck"), I think it should be acknowledged.
- Variants of transformers with competition among specialists via sparsity have also been proposed, e.g. adaptively sparse transformers [2]. That framework is an alternative to top-k softmax used in this paper.
- Empirical studies which analyze the redundancy among specialists (in this case attention heads) and propose strategies to prune them have also been made by [3].

[1] https://arxiv.org/abs/1902.09113
[2] https://arxiv.org/abs/1909.00015
[3] https://arxiv.org/abs/1905.09418

Minor point: "Hence unlike pairwise interaction, messages passed among neural modules in the shared workspace setting also include HO interaction terms" -- I believe higher-order interaction happens too every two layers with pairwise interaction. Perhaps this should be clarified.